# A mechanistic insight into sources of error of visual working memory in multiple sclerosis

Ali Motahharynia[1,2†], Ahmad Pourmohammadi[1,2,3†], Armin Adibi[1,2], Vahid Shaygannejad[1,2,4], Fereshteh Ashtari[1,2,4], Iman Adibi[1,2,4*], Mehdi Sanayei[1,2,3*]

[1]Center for Translational Neuroscience, Isfahan University of Medical Sciences, Isfahan, Islamic Republic of Iran; [2]Isfahan Neuroscience Research Center, Isfahan University of Medical Sciences, Isfahan, Islamic Republic of Iran; [3]School of Cognitive Sciences, Institute for Research in Fundamental Sciences (IPM), Tehran, Islamic Republic of Iran; [4]Department of Neurology, School of Medicine, Isfahan University of Medical Sciences, Isfahan, Islamic Republic of Iran

**\*For correspondence:**
i.adibi@gmail.com (IA);
mehdi.sanayei@gmail.com (MS)

†These authors contributed equally to this work

**Competing interest:** The authors declare that no competing interests exist.

**Abstract** Working memory (WM) is one of the most affected cognitive domains in multiple sclerosis (MS), which is mainly studied by the previously established binary model for information storage (slot model). However, recent observations based on the continuous reproduction paradigms have shown that assuming dynamic allocation of WM resources (resource model) instead of the binary hypothesis will give more accurate predictions in WM assessment. Moreover, continuous reproduction paradigms allow for assessing the distribution of error in recalling information, providing new insights into the organization of the WM system. Hence, by utilizing two continuous reproduction paradigms, memory-guided localization (MGL) and analog recall task with sequential presentation, we investigated WM dysfunction in MS. Our results demonstrated an overall increase in recall error and decreased recall precision in MS. While sequential paradigms were better in distinguishing healthy control from relapsing-remitting MS, MGL were more accurate in discriminating MS subtypes (relapsing-remitting from secondary progressive), providing evidence about the underlying mechanisms of WM deficit in progressive states of the disease. Furthermore, computational modeling of the results from the sequential paradigm determined that imprecision in decoding information and swap error (mistakenly reporting the feature of other presented items) was responsible for WM dysfunction in MS. Overall, this study offered a sensitive measure for assessing WM deficit and provided new insight into the organization of the WM system in MS population.

## eLife assessment

This paper provides **valuable** information regarding visuospatial working memory performance in patients with MS compared to healthy controls, using a relatively novel continuous measure of visual working memory. There are some weaknesses with the way the clinical groups were matched, but those limitations are acknowledged and the strength of evidence for the claims is otherwise **convincing**. The paper will be of interest to those working in the field of clinical neuroscience.

## Introduction

Multiple sclerosis (MS) is a debilitating inflammatory disorder characterized by demyelinating central nervous system (CNS) plaques creating a progressive neurodegenerative state with heterogeneous

**eLife digest** Working memory is a system that temporarily stores and manipulates information used in tasks like decision-making and reasoning. Patients with multiple sclerosis – a condition that can affect the brain and spinal cord – often have impaired working memory, which can negatively affect their quality of life.

Traditionally, working memory has been evaluated using tests that determine whether a patient can recall an item or not. In this approach, an incorrect response implies a complete absence of information regarding the specific item, resulting in a binary evaluation. More recently, researchers have shown that the precision of the memories people recall degrades gradually as they are asked to remember more things and that focusing on an item negatively affects recall precision for other items. This implies that working memory is reorganised flexibly between memorised items, a so-called 'resource model'.

Unlike previous research, which favoured a binary model, Motahharynia et al. used a resource model to study visual working memory impairment in multiple sclerosis. The study participants consisted of healthy volunteers and patients with two subtypes of multiple sclerosis. Each participant completed one of two different types of test. In one, they were shown targets for short periods of time and then asked to pinpoint their position after they disappeared. In the other, participants were asked to memorise the orientation and colour of consecutively presented bars.

The findings confirmed that multiple sclerosis patients had worse memory recall than people without the disease. However, computer modelling provided insights into the sources of error in working memory dysfunction, showing that the memory deficiency was due to imprecision in recalling information and 'swap errors', the phenomenon of mistakenly reporting the property of other memorised items. This rise in swap errors is likely due to an increase in unwanted signals, or noise, in the brains of multiple sclerosis patients.

Motahharynia et al. have presented a sensitive way of measuring working memory deficiency. Importantly, the measurements were able to distinguish between different stages of multiple sclerosis. This could help doctors detect disease progression earlier, allowing for more timely and effective treatment interventions. This method could also be useful in the development and testing of drugs for therapy.

---

clinical characteristics (*Benedict et al., 2020*; *Dobson and Giovannoni, 2019*). Impairment in cognitive function is a common clinical manifestation of MS, which detrimentally affects different aspects of patients' daily life, from decreased physical performance and productivity to unemployment (*Benedict et al., 2020*; *Clemens and Langdon, 2018*). One of the frequently affected domains of cognition in MS is working memory (WM) (*Vacchi et al., 2017*), which, due to its essential role in several cognitive processes (*Miller et al., 2018*; *Nee and D'Esposito, 2016*), is one of the main areas of MS research (*Costers et al., 2021*; *Covey et al., 2011*; *Hulst et al., 2017*; *Rocca et al., 2014*; *Vacchi et al., 2017*).

Multiple neuropsychological cognitive paradigms, such as paced auditory serial addition test (PASAT), n-back, and delayed-match to sample, were developed to investigate different aspects of WM deficit in MS (*Costers et al., 2021*; *Parmenter et al., 2006*; *Pourmohammadi et al., 2023*; *Rocca et al., 2014*; *Stojanovic-Radic et al., 2015*; *Vacchi et al., 2017*). The basis of these change detection paradigms is the slot model of WM (*Ma et al., 2014*). In this quantized model, WM is considered a short-term storage for a limited number of items (*Cowan, 2001*; *Ma et al., 2014*; *Miller, 1956*), storing the information in a binary format. This assumption creates an all-or-none condition in which only the stored items in these limited slots will be remembered (*Ma et al., 2014*). Nonetheless, recent observations from analog recall paradigms assessing the precision of WM determined dynamic allocation of WM resources (*Bays et al., 2009*; *Bays and Husain, 2008*; *Gorgoraptis et al., 2011*; *Schneegans and Bays, 2016*). Each stored item in this framework possesses a fraction of WM storage in which the allocated space changes dynamically between them (*Bays and Husain, 2008*; *Ma et al., 2014*; *Schneegans and Bays, 2016*). This concept is the foundation for the resource-based model of WM.

The analog nature of inputs in resource-based model paradigms makes it possible to investigate the resolution and variability of stored memory (*Peich et al., 2013*; *Zokaei et al., 2015*). Also, assessing

the distribution of error in these paradigms further helped in uncovering the underlying structure of the visual WM system (*Liang et al., 2016*; *Lugtmeijer et al., 2021*; *McMaster et al., 2022*; *Peich et al., 2013*; *Zokaei et al., 2020*). An analog recall task is a paradigm in which subjects need to simultaneously recall multiple features of items in a continuous space, hence requiring encoding the information of connected features in addition to their distinct value (e.g., object, location, and object-location binding information) (*McMaster et al., 2022*; *Peich et al., 2013*; *Zokaei et al., 2020*). According to the study of Bays et al., there are three different sources of error for recalling information in visual WM tasks with connected features (*Bays et al., 2009*). They are identified as (i) the Gaussian variability in response around the target value (target response proportion), (ii) Gaussian variability in response around the non-target value, that is, mistakenly reporting feature of other presented items (swap error), and (iii) random responses (uniform response proportion) (*Bays et al., 2009*; *Zokaei et al., 2020*).

Studies based on analog recall paradigms unraveled new insights into the sources of recall error in neurodegenerative disorders. It was determined that random response and swap error contribute to the impairment of visual WM in Parkinson's and Alzheimer's diseases, respectively (*Liang et al., 2016*; *Zokaei et al., 2020*). However, regardless of the influential impact of these study designs on the discovery of novel mechanistic insights into the organization of visual WM system, it has not gotten enough attention in the field of MS research.

This study followed our previous study in which the quantity of MS-related visual WM was assessed based on the slot model (*Pourmohammadi et al., 2023*). Here, we aimed to evaluate recall precision (quality) based on the resource-based model paradigms. In this regard, we developed two analog recall paradigms, a memory-guided localization (MGL) and two analog recall tasks with sequential bar presentation (3 bar and 1 bar). Primarily using the simplistic design of MGL, recall error (absolute error) and precision of visual WM were assessed. Similarly, recall error and precision were evaluated using the two designed analog recall paradigms with sequential presentation, that is, the low memory load, 1 bar, and high memory load, 3 bar conditions, respectively. Furthermore, the classifying performance of these paradigms in distinguishing different groups was assessed. Finally, the distribution of recall error in analog paradigms with sequential presentation was further investigated using the probabilistic model of Bays, Catalao, and Husain (*Bays et al., 2009*). In both MGL and sequential paradigms (low and high memory load conditions), recall error and precision were impaired in MS. The dissociable function of these paradigms in classifying MS subtypes (relapsing-remitting and secondary progressive) gave some clues about the underlying structure of WM deficit in progressive states of the disease. Investigation into sources of error in the high memory load condition revealed that target response proportion and swap error (non-target response proportion) contribute to visual WM dysfunction in MS.

## Results

In the MGL paradigm (*Figure 1A*), 45patients (19 relapsing-remitting MS [RRMS] and 26 secondary progressive MS [SPMS]) and 24 healthy controls participated. As mentioned in the 'Materials and methods' section, the mean data of five participants (one healthy control, three RRMS, and one SPMS) were excluded from further analysis. In the sequential paradigms (*Figure 1B* and C), from a total of 76patients (42 RRMS and 34 SPMS) and 49 healthy controls who participated, three healthy controls, three RRMS, and two SPMS were excluded. The demographic and clinical data of participants are summarized in *Tables 1 and 2*.

### Recall error in multiple sclerosis

In the MGL paradigm (*Figure 1A*), recall error was evaluated as a function of distance using a mixed-model ANOVA. Recall error was significantly different between groups ($F(2,61) = 14.57$, $p<10^{-5}$) and distances ($F(2,61) = 85.03$, $p<10^{-23}$, *Figure 2A*). A significant interaction was also observed between group and distance ($F(4,61) = 7.24$, $p<10^{-4}$). Tukey post hoc test determined that recall error was significantly higher in SPMS (1.86° ± 0.92° visual degree) compared to healthy control (0.97 ± 0.26, $p<10^{-4}$) and RRMS (1.09 ± 0.27, $p<10^{-3}$). No significant difference was detected between RRMS and healthy control (p=0.83). Similarly, recall error as a function of delay interval was also evaluated. Recall error was significantly different between delay intervals ($F(4,61) = 18.89$, $p<10^{-12}$, *Figure 2D*). No



**Figure 1.** Schematic design of visual working memory (WM) paradigms. (**A**) In the memory-guided localization (MGL) paradigm, participants were asked to memorize and then localize the position of the target circle following a random delay interval of 0.5, 1, 2, 4, or 8 s. Following their response, visual feedback was presented. (**B**) In the sequential paradigm with 3 bar (high memory load condition), a sequence of three colored bars was presented consecutively. Participants were asked to match the orientation of the probe bar to the previously presented bar with the same color. Visual feedback was displayed following their response. (**C**) The 1 bar paradigm (low memory load condition) has the same structure as the 3 bar paradigm except for presenting one bar instead of three.

significant interaction was observed between group and delay interval [$F(8,61) = 0.69$, p=0.70]. After adjusting for cognitive ability, assessed by the Montreal Cognitive Assessment (MoCA) screening tool (cognitive performance was significantly different between groups), the effect of group on recall error remained significant (***Supplementary file 1***).

While reaction time (RT) was significantly different between groups ($F(2,61) = 26.44$, p<$10^{-8}$) and distances ($F(2,61) = 25.94$, p<$10^{-9}$, ***Figure 2C***), it was not significantly different between delay intervals ($F(4,61) = 0.97$, p=0.43, ***Figure 2F***). No significant interaction was observed between group and distance ($F(4,61) = 2.06$, p=0.09) or group and delay interval ($F(8,61) = 0.86$, p=0.55). The statistical results of RT are summarized in ***Supplementary file 2***.

Recall error was evaluated for sequential tasks using the same method. In the 'high memory load' condition (i.e., sequential paradigm with 3 bar, ***Figure 1B***), recall error was significantly different between groups ($F(2,114) = 28.18$, p<$10^{-9}$) and bar orders ($F(2,114) = 48.74$, p<$10^{-17}$, ***Figure 2G***). No significant interaction was observed between them ($F(4,114) = 1.21$, p=0.31). Tukey post hoc test showed that recall error was significantly higher in RRMS (0.68 ± 0.20 radian) and SPMS (0.76 ± 0.17) compared to healthy control (0.48 ± 0.16, p<$10^{-5}$, p<$10^{-8}$, respectively). However, no significant difference was detected between RRMS and SPMS groups (p=0.14). After adjusting for gender, age, education, and cognitive ability (they were significantly different between groups), the group's effect on recall error remained significant (***Supplementary file 3***). Moreover, while RT was also significantly

**Table 1.** Demographic and clinical profiles of participants in the MGL paradigm.

| | HC (n = 23) | RRMS (n = 16) | SPMS (n = 25) | p |
|---|---|---|---|---|
| Gender (F:M) | 13:10 | 14:2 | 17:8 | 0.12 |
| Age (year) | 35.9 ± 8.34 | 37.25 ± 6.63 | 39.28 ± 5.56 | 0.25 |
| Education (years) | 13.30 ± 2.74 | 13.69 ± 3.34 | 13.56 ± 3.22 | 0.86 |
| Cognitive ability[†] (NL:MCI) | 23:0 | 14:2 | 19:6 | <0.05* |
| Disease duration (years) | N/A | 8.562 ± 3.20 | 11.56 ± 3.28 | <0.02* |
| EDSS | N/A | 1.28 ± 0.79 | 2.740 ± 1.23 | <0.0002* |
| DMT (platform: non-platform) | N/A | 2:14 | 0:25 | 0.07 |

All data, except for the categorical information, are presented as mean ± standard deviation.

Gender, cognitive ability, and DMT were compared using the chi-square test. Age and education were compared using one-way ANOVA and Kruskal–Wallis $H$ test, respectively. Disease duration and EDSS score were compared using Mann–Whitney $U$ test and independent-sample $t$-test, respectively.

HC = healthy control, RRMS = relapsing-remitting multiple sclerosis, SPMS = secondary progressive multiple sclerosis, NL = normal (MoCA score ≥26), MCI = mild cognitive impairment (MoCA score = 18–25), EDSS = expanded disability status scale, DMT = disease-modifying therapy, platform treatment = interferon β-1a and glatiramer acetate, non-platform treatment = rituximab, ocrelizumab, fingolimod, dimethyl fumarate, and natalizumab, N/A = not applicable.

*p<0.05.

[†]Assessed based on the Montreal Cognitive Assessment (MoCA) test classification.

The online version of this article includes the following source data for table 1:

**Source data 1.** Clinical and demographic data of participants in the memory-guided localization (MGL) paradigm.

**Table 2.** Demographic and clinical profiles of participants in the sequential paradigms.

| | HC (n = 46) | RRMS (n = 39) | SPMS (n = 32) | p |
|---|---|---|---|---|
| Gender (F:M) | 16:30 | 23:16 | 22:10 | <0.008* |
| Age (year) | 30.5 ± 10.37 | 32.03 ± 6.72 | 39.00 ± 6.43 | <10^{-6}* |
| Education (years) | 16.95 ± 2.23 | 13.87 ± 3.41 | 13.67 ± 2.73 | <10^{-7}* |
| Cognitive ability[†] (NL: MCI) | 42:4 | 28:11 | 18:13 | <0.003* |
| Disease duration (years) | N/A | 6.60 ± 3.84 | 9.37 ± 4.43 | <0.007* |
| EDSS | N/A | 1.49 ± 1.01 | 3.86 ± 1.74 | <10^{-7}* |
| DMT (platform: non-platform) | N/A | 5:34 | 1:31 | 0.14 |

All data, except for the categorical information, are presented as mean ± standard deviation.

One MoCA value in the SPMS group is missing. Gender, cognitive ability, and DMT were compared using the chi-square test. Age and education were compared using the Kruskal–Wallis $H$ test. Disease duration and EDSS score were compared using an independent-sample $t$-test and Mann–Whitney $U$ test, respectively. Dunn's post hoc test was performed following significant results of age and education, and the adjusted p-value following Bonferroni correction for multiple tests are reported: Age: healthy vs. RRMS: p=0.27, healthy vs. SPMS: $p<10^{-6}$*, and RRMS vs. SPMS: p<0.005*. Education: healthy vs. RRMS: $p<10^{-5}$*, healthy vs. SPMS: $p<10^{-5}$*, and RRMS vs. SPMS: p=1.

HC = healthy control, RRMS = relapsing-remitting multiple sclerosis, SPMS = secondary progressive multiple sclerosis, NL = normal (MoCA score ≥26), MCI = mild cognitive impairment (MoCA score = 18–25), EDSS = expanded disability status scale, DMT = disease-modifying therapy, platform treatment = interferon β-1a and glatiramer acetate, non-platform treatment = rituximab, ocrelizumab, fingolimod, dimethyl fumarate, and natalizumab, N/A = ot applicable.

*p<0.05.

[†]Assessed based on the Montreal Cognitive Assessment (MoCA) test classification.

The online version of this article includes the following source data for table 2:

**Source data 1.** Clinical and demographic data of participants in the sequential paradigms.

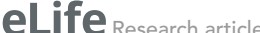

**Figure 2.** Recall error and precision of healthy control and multiple sclerosis (MS) subtypes (relapsing-remitting [RRMS] and secondary progressive [SPMS]) in visual working memory (WM) paradigms. (**A**) Recall error, (**B**) recall precision, and (**C**) reaction time as a function of distance for the memory-guided localization (MGL) paradigm. (**D–F**) The same as a function of delay interval. (**G**) Recall error, (**H**) recall precision, and (**I**) reaction time as a function of bar order in the sequential paradigms with 3 bar (left of each subplot) and 1 bar (right of each subplot). Data are represented as mean ± SEM.

*Figure 2 continued on next page*

*Figure 2 continued*

The online version of this article includes the following source data for figure 2:

**Source data 1.** Statistical reports corresponding to *Figure 2*.

different between groups ($F(2,114) = 12.95$, $p<10^{-5}$) and bar orders ($F(2,114) = 5.92$, $p<0.004$, *Figure 2I*), no significant interaction was observed between group and bar order ($F(4,114) = 0.16$, $p=0.96$). The statistical results of RT are summarized in *Supplementary file 4*.

In the 'low memory load' condition (i.e., 1 bar paradigm, *Figure 1C*), recall error was significantly different between groups ($F(2,114) = 36.85$, $p<10^{-12}$, *Figure 2G*). Tukey post hoc test showed that recall error was significantly higher in RRMS ($0.33 \pm 0.12$ radian) and SPMS ($0.43 \pm 0.12$) compared to healthy control ($0.22 \pm 0.08$, $p<10^{-4}$, $p<10^{-8}$, respectively). We also observed a significant difference between RRMS and SPMS ($p<10^{-3}$). After adjusting for gender, age, education, and cognitive ability, the group's effect on recall error remained significant (*Supplementary file 5*). Correspondingly, RT differed significantly between groups ($F(2,114) = 12.59$, $p<10^{-4}$, *Figure 2I*). The statistical results of RT are summarized in *Supplementary file 4*.

## Recall precision in multiple sclerosis

In the MGL paradigm, recall precision was also significantly different between groups ($F(2,61) = 13.74$, $p<10^{-4}$) and distances ($F(2,61) = 23.39$, $p<10^{-8}$, *Figure 2B*). No significant interaction was observed between group and distance ($F(4,61) = 0.91$, $p=0.46$). Post hoc analysis determined that recall precision was significantly lower in SPMS ($0.87 \pm 0.52/°$) than in both RRMS ($1.32 \pm 0.51$, $p<0.039$) and healthy control ($1.70 \pm 0.60$, $p<10^{-5}$). Recall precision was not significantly different between RRMS and healthy control ($p=0.08$). Moreover, this effect remained significant after adjusting for cognitive ability (*Supplementary file 1*). We also determined the effects of delay interval on recall precision. In this analysis, groups ($F(2,61) = 14.57$, $p<10^{-5}$), delay intervals ($F(4,61) = 7.30$, $p<10^{-4}$), and their interaction ($F(8,61) = 2.44$. $p<0.02$) had significant effects on recall precision (*Figure 2E*). Post hoc analysis showed the same pattern of result (SPMS = $0.84 \pm 0.47$; RRMS = $1.39 \pm 0.46$; healthy = $1.63 \pm 0.61$; SPMS vs. RRMS: $p<0.005$; SPMS vs. healthy: $p<10^{-5}$, RRMS vs. healthy: $p=0.34$). This effect remained significant after adjusting for cognitive ability (*Supplementary file 1*).

Similarly, in the high memory load condition, recall precision was significantly different between groups ($F(2,114) = 25.23$, $p<10^{-9}$, *Figure 2H*) and bar orders ($F(2,114) = 20.70$, $p<10^{-8}$). However, no significant interaction was observed between them ($F(4,114) = 1.84$, $p=0.12$). Besides, while in post hoc analysis recall precision was significantly higher in healthy control ($2.39 \pm 0.78/$radian) than in both RRMS ($1.68 \pm 0.50$, $p<10^{-6}$) and SPMS patients ($1.52 \pm 0.31$, $p<10^{-7}$), no difference was observed between RRMS and SPMS participants ($p=0.49$). After adjusting for gender, age, education, and cognitive ability, the effect of group on recall precision remained significant (*Supplementary file 3*).

Accordingly, the low memory load condition showed the same pattern. Recall precision significantly differed between groups ($F(2,114) = 25.48$, $p<10^{-9}$, *Figure 2H*). Post hoc analysis determined that recall precision was significantly higher in healthy control [$6.10 \pm 2.41/$radian] than in RRMS ($4.16 \pm 1.98$, $p<10^{-4}$) and SPMS ($2.95 \pm 1.05$, $p<10^{-8}$). Moreover, there was a significant difference between RRMS and SPMS ($p<0.031$), which, after adjusting for gender, age, education, and cognitive ability, remained significant (*Supplementary file 5*).

## Distribution of error in recalling information in multiple sclerosis

The distribution of error in recalling information was assessed further to investigate the underlying mechanisms of WM impairment in MS. In this regard, the recorded data from the sequential paradigms was fitted to a probabilistic model developed by *Bays et al., 2009*. In line with the results of recall error and precision, for the high memory load condition, circular standard deviation (SD) of von Mises distribution of recall error was significantly different between groups ($F(2,114) = 26.79$, $p<10^{-9}$) and bar orders ($F(2,114) = 14.95$, $p<10^{-6}$, *Figure 3A*). At the same time, they had no significant interaction ($F(4,114) = 1.19$, $p=0.31$). von Mises SD of recall error was lower in healthy control ($0.51 \pm 0.12$) than both RRMS ($0.69 \pm 0.20$, $p<10^{-5}$) and SPMS ($0.76 \pm 0.15$, $p<10^{-8}$). There was no difference between RRMS and SPMS in the von Mises SD parameter ($p=0.16$). In the low memory load condition, von Mises SD was affected by groups ($F(2,114) = 33.07$, $p<10^{-11}$, *Figure 3A*). Again, von



**Figure 3.** Sources of recall error in high and low memory load conditions (3 bar and 1 bar, respectively). (**A**) von Mises SD (circular standard deviation of von Mises distribution), (**B**) Target response (probability of response around the target value), (**C**) swap error (probability of response around the non-target values), and (**D**) uniform response (probability of random response) for healthy control and multiple sclerosis (MS) subtypes in the sequential paradigms with 3 bar (left of each subplot) and 1 bar (right of each subplot). Data are represented as mean ± SEM.

The online version of this article includes the following source data and figure supplement(s) for figure 3:

**Source data 1.** Statistical reports corresponding to *Figure 3* and *Figure 3—figure supplement 1*.

**Figure supplement 1.** Isolated effect of orientation in the high and low memory load conditions.

Mises SD was lower in healthy control (0.25 ± 0.10) than both RRMS (0.38 ± 0.14, $p<10^{-5}$) and SPMS (0.47 ± 0.12, $p<10^{-8}$). There was a significant difference between RRMS and SPMS in von Mises SD ($p<0.02$). After adjusting for gender, age, education, and cognitive ability, the group's effect on von Mises SD remained significant in both high and low memory load conditions (*Supplementary file 3* and *Supplementary file 5*).

According to the study by *Bays et al., 2009*, there are three sources of error for recalling information. The sources of these errors were defined as Gaussian variability in response around (i) target (target response proportion, *Figure 3B*) and (ii) non-target values (swap error, *Figure 3C*) and (iii) random responses (uniform response proportion, *Figure 3D*). In the high memory load condition, target response proportion was significantly different between groups ($F(2,114) = 11.04$, $p<10^{-4}$) and bar orders ($F(2,114) = 10$, $p<10^{-4}$, *Figure 3B*). No significant interaction was observed between group and bar order ($F(4,114) = 0.43$, $p=0.78$). Target proportion was higher in healthy control ($0.88 \pm 0.11$) than both RRMS ($0.79 \pm 0.12$, $p<0.003$) and SPMS ($0.76 \pm 0.14$, $p<10^{-4}$). There was no significant difference in target proportion between RRMS and SPMS groups ($p=0.54$). After adjusting for gender, age, years of education, and cognitive ability, the effect of group on target proportion remained significant (*Supplementary file 3*). Moreover, the nearest-neighbor analysis was performed to further evaluate the effect of target proportion in the absence of swap error. Removing the effect of swap error allowed us to assess the isolated effect of orientation recall. The findings from the nearest-neighbor analysis showed a similar pattern of results (*Figure 3—figure supplement 1*). The isolated effect of orientation was significantly different between groups ($F(2,114) = 29.26$, $p<10^{-10}$) among different bar orders ($F(2,114) = 7.07$, $p<0.002$). No significant interaction was observed between group and bar order ($F(4,114) = 0.58$, $p=0.67$). After adjusting for gender, age, education, and cognitive ability, the results of nearest-neighbor analysis remained significant (*Supplementary file 3*).

In the low memory load condition, group significantly affected target response proportion ($F(2,114) = 3.11$, $p<0.049$, *Figure 3B*). While target proportion in the healthy group ($0.98 \pm 0.03$) was significantly higher than SPMS patients ($0.94 \pm 0.09$, $p<0.04$), after adjusting for gender, age, years of education, and cognitive ability, this effect became insignificant (*Supplementary file 5*). In addition, the target proportion did not significantly differ between healthy vs. RRMS ($0.96 \pm 0.08$, $p=0.48$) and RRMS vs. SPMS population ($p=0.37$).

In line with the above findings, swap error was higher in the MS population. In the high memory load condition, swap error was significantly different between groups ($F(2,114) = 7.11$, $p<0.002$) and bar orders ($F(2,114) = 31.05$, $p<10^{-11}$, *Figure 3C*). No significant interaction was observed between group and bar order ($F(4,114) = 1.45$, $p=0.22$). Swap error was lower in healthy control ($0.07 \pm 0.06$) than in both RRMS ($0.11 \pm 0.09$, $p<0.05$) and SPMS patients ($0.14 \pm 0.09$, $p<0.002$). There was no significant difference in swap error between RRMS and SPMS ($p=0.41$). After adjusting for gender, age, years of education, and cognitive ability, the group's effect on swap error remained significant (*Supplementary file 3*). Moreover, while in the high memory load condition the uniform response proportion was different between groups ($F(2,114) = 5.50$, $p<0.006$, *Figure 3D*), no such differences were observed between bar orders ($F(2,114) = 0.81$, $p=0.45$) or the interaction between them ($F(4,114) = 0.18$, $p=0.95$). Post hoc analysis revealed that uniform proportion was lower in healthy control ($0.05 \pm 0.08$) than both RRMS ($0.09 \pm 0.08$, $p<0.03$) and SPMS ($0.10 \pm 0.08$, $p<0.02$). Moreover, there was no significant difference in uniform response proportion between RRMS and SPMS groups ($p=0.95$). Additionally, although after adjusting for gender, the effect of group on uniform proportion remained significant, adding age, years of education, or cognitive ability made this effect insignificant (*Supplementary file 3*). The result of uniform response proportion in the low memory load condition is mathematically same as the target proportion (uniform proportion = 1 – target proportion, since there was no swap error in the 1 bar condition).

## Dissociable function of MGL and sequential paradigms

The classifying ability of MGL and sequential paradigms in differentiating healthy control from MS patients was assessed based on recall error parameters using the receiver operating characteristic (ROC) analysis (*Figure 4A–C*). The accuracy of MGL and sequential paradigms with 3 bar and 1 bar in differentiating MS patients from healthy participants was 80% (*Figure 4A*), 83.4% (*Figure 4B*), and 86.2% (*Figure 4C*), respectively. A closer look at *Figure 2A, D and G* suggested that these paradigms differentiate MS and healthy control with distinct patterns. Hence, we separately applied ROC analysis to healthy control vs. RRMS, healthy control vs. SPMS, and RRMS vs. SPMS for MGL (*Figure 4D*) and sequential paradigms with 3 bar (*Figure 4E*) and 1 bar (*Figure 4F*). While the MGL paradigm had good performance in differentiating SPMS from healthy control (90.1%) and SPMS from RRMS (84.7%), it had poor ability in distinguishing healthy control from RRMS (64.1%, *Figure 4D*). Accordingly, although the 3 bar paradigm also had good accuracy in differentiating healthy control from



**Figure 4.** Classifying performance of visual working memory (WM) paradigms in differentiating healthy control from multiple sclerosis (MS) and MS subtypes, and MS subtypes from each other. Receiver operating characteristic (ROC) curve demonstrated the accuracy of (**A**) memory-guided localization (MGL) and sequential paradigms with (**B**) 3 bar and (**C**) 1 bar in distinguishing healthy control from MS patients. The precision of these paradigms in dissociating healthy control from MS subtypes (relapsing-remitting MS [RRMS] and secondary progressive MS [SPMS]) and MS subtypes from each other is represented as the area under the curve (AUC) for (**D**) MGL and sequential paradigms with (**E**) 3 bar and (**F**) 1 bar.

The online version of this article includes the following source data for figure 4:

**Source data 1.** Statistical reports corresponding to *Figure 4*.

SPMS (88.4%) and better results (compared to MGL) in discriminating healthy control from RRMS (79.3%), it did a poor job in discriminating MS subtypes (62%, *Figure 4E*). Complementary to the above findings, the 1 bar paradigm showed an in-between pattern of results. The 1 bar paradigm accurately discriminates healthy control from SPMS (94.4%), while it also performed well in differentiating healthy control from RRMS (79.6%). However, compared to MGL, it had a weaker ability to discriminate MS subtypes (72.3%, *Figure 4F*).

## Discussion

In this study, we investigated the visual WM deficits in MS using two continuous reproduction paradigms: MGL and analog recall paradigms with sequential presentation. Our results align with the

previous reports regarding WM deficit in MS (*Costers et al., 2021*; *Parmenter et al., 2006*; *Pourmo-hammadi et al., 2023*; *Stojanovic-Radic et al., 2015*; *Vacchi et al., 2017*). Complementary to these reports, which envisaged a binary model for storing information (slot-based model), we assessed recall precision in WM using analog reproduction paradigms (resource-based model). Further, by utilizing the unique design of our sequential paradigm, which allows us to assess the distribution of error in recalling information, we introduced a new mechanistic insight into the visual WM dysfunction in MS.

Although the results from both MGL and sequential paradigms showed an overall decrease in recall precision and increased recall error in MS population, the post hoc analysis demonstrated inconsistent results. In the MGL paradigm, while SPMS patients performed worse than other groups, no significant difference was observed between healthy control and RRMS. This result contrasted with the sequential paradigm with 3 bar (high memory load condition), in which MS subtypes (RRMS vs. SPMS) were not significantly different, and healthy control performed better than RRMS and SPMS. The situation also varied for the 1 bar paradigm (low memory load condition), in which all three groups performed with different levels of precision. ROC analysis further confirmed these results, which determined a dissociation between the classifying performance of MGL and sequential paradigms. Additionally, although the low memory load condition was better than the high memory load condition in distinguishing MS subtypes, its classifying performance was not as good as the MGL paradigm. It seems that these paradigms evaluated distinct aspects of WM dysfunction in MS.

The dissociable function of our paradigms could arise from using different types of stimuli (location in MGL vs. orientation in sequential paradigms), in which spatial WM was assessed in the MGL paradigm. As the spatial WM process was associated with the function of the hippocampus (*Rolls, 2018*), the observed difference could indicate more hippocampal disruption in SPMS patients. This finding is in line with previous studies that showed more hippocampal regional loss (*Sicotte et al., 2008*) and increased hippocampus neuroinflammatory activity in SPMS compared to RRMS (*Cree et al., 2021*), suggesting that assessment of the spatial WM could be a specific marker for disorganization of the WM system in SPMS.

Another explanation is the long delay interval in MGL, which assessed the maintenance of information. Therefore, the observed difference could be due to additional impairment of SPMS patients in keeping that information. This finding is in line with our previous study, which showed that change detection paradigms with long delay intervals were promising in differentiating MS subtypes (*Pourmohammadi et al., 2023*). At the same time, one may debate that this difference was related to the longer stimulus presentation time in the MGL paradigm (1000 ms vs. 500 ms in the sequential paradigm). However, since the stimulus presentation time was adequate in sequential paradigms and the stimuli were presented consecutively, it did not seem that the inadequate time for encoding information was responsible for this difference (*Peich et al., 2013*; *Zokaei et al., 2015*).

Additionally, one may argue that the observed dissociation could be due to the extra binding process needed in the sequential paradigm with 3 bar. However, the results from the low memory load condition and our findings from the sequential paradigm nearest-neighbor analysis, which provided a proxy to assess the isolated effect of orientation, demonstrated a similar pattern of dissociation in the absence of a binding effect. Based on these findings, we concluded that the binding process was not responsible for the observed dissociation. Nevertheless, since the evaluated binding process was an intra-term association (i.e., conjunctive binding), we could not be assured that the same results would be reached for an inter-term association (i.e., relational binding) (*Parra et al., 2015*). This issue becomes more interesting when we realize that the relational binding function is mainly centered on the hippocampus (*Liang et al., 2016*; *Parra et al., 2015*), the structure we presumed was responsible for the observed dissociation in the MGL paradigm.

Finally, due to the diffuse pattern of involved brain areas in MS and evidence demonstrating that brain networks accounted for different WM processes, it is reasonable to assume that distinct WM-related networks, instead of a single region, were responsible for the observed dissociable patterns (*Bastin et al., 2019*; *Dobson and Giovannoni, 2019*; *Figueroa-Vargas et al., 2020*; *Lugt-meijer et al., 2021*; *Vacchi et al., 2017*).

We applied a probabilistic model to disentangle the error distribution in recalling information in the sequential paradigms to further investigate the underlying mechanism behind the visual WM dysfunction in MS. Our finding revealed that in addition to imprecision in decoding information, swap error (mistakenly reporting a non-target feature) contributed to WM dysfunction in individuals with

MS. Swap error has been associated with various mechanisms, including the variability in cue feature dimension, cue-independent sources, and strategic guessing (*McMaster et al., 2022*). The recent study by McMaster et al. comprehensively investigated these hypotheses and determined that the variability in cue feature dimension could solely account for the swap error mechanism (*McMaster et al., 2022*). The neural correlates of swap error could be explained by the imprecision in decoding information in the cue feature dimension (i.e., color in this study) due to the noise in neural activity. Hence, a non-cued item may mistakenly be considered as the actual cue, leading to the reporting of the corresponding non-target feature as a response (*Schneegans and Bays, 2017*). Therefore, the swap error observed in this study can be attributed to the variability in the color dimension. The neural underpinning of this observation in MS population could be related to their impaired neural activity. Recent studies have demonstrated disturbed neural activity during WM tasks in MS (*Costers et al., 2021*; *Figueroa-Vargas et al., 2020*). For instance, Costers et al. showed a disturbed theta band oscillation, which plays a crucial role in WM encoding information, in the hippocampal area of MS population (*Costers et al., 2021*). While this evidence provides a primary insight into the possible underlying mechanisms of WM dysfunction in MS, future electrophysiological studies using the cued recall paradigms could further elucidate this matter.

Swap errors (misbinding errors) have been observed in various neurological disorders, including different types of Alzheimer's disease (*Cecchini et al., 2023*; *Della Sala et al., 2012*; *Liang et al., 2016*; *Zokaei et al., 2020*), epileptic patients with temporal lobe lobectomy (*Zokaei et al., 2019*), and voltage-gated potassium channel complex antibody (VGKC-Ab) limbic encephalitis (*Pertzov et al., 2013*). From a neuroanatomical perspective, previous studies proposed that impairment in hippocampus and medial temporal lobe regions contributes to problems in relational binding (*Della Sala et al., 2012*; *Liang et al., 2016*; *Pertzov et al., 2013*; *Zokaei et al., 2019*). Furthermore, whereas regions related to conjunctive binding have yet to be better understood, occipital and parietal regions seem to be mainly related to conjunctive processing (*Cecchini et al., 2023*). Additionally, a recent study by Valdés Hernández and colleagues also demonstrated the possible role of globus pallidus in conjunctive binding (*Valdés Hernández et al., 2020*). On the other hand, a study by Vacchi et al. demonstrated less activation of the superior and inferior parietal lobule during an N-back fMRI task in individuals with MS (*Vacchi et al., 2017*). Moreover, globus pallidus was shown to be involved in MS (*Fujiwara et al., 2017*). Hence, considering the convergence of brain regions involved in conjunctive processing (assessed through the current sequential paradigm) with those shown to be affected in MS, the parietal regions and globus pallidus could be the candidate regions responsible for the observed impairment. Additionally, based on the evidence showing the involvement of hippocampal regions in MS (*Rocca et al., 2018*; *Sicotte et al., 2008*), we also expect to see relational processing impairment; however, this study's design did not allow us to evaluate this condition. Further neuroimaging studies are needed to validate these findings. Eventually, the insignificant results from the low memory load condition suggested more impairment in visual WM under high memory load situations, which was not unexpected.

Despite these findings, our study had some limitations that should be addressed in future research. This study only assessed WM dysfunction using behavioral paradigms. Further structural and functional evaluations should be performed to confirm our suggested brain areas associated with conjunctive binding and spatial WM deficit in MS. Concurrent assessment of brain networks using fMRI and EEG or volumetric studies alongside behavioral paradigms could address this issue. Furthermore, although we hypothesized that relational binding could be a specific marker for the progressive state of the disease based on the more disruption of hippocampal-related areas in SPMS, this study's design did not allow us to evaluate this assumption. Future studies assessing the source of WM deficit regarding relational binding could elucidate this statement. Additionally, although clinical and cognitive assessments were performed to mitigate the possible confounding effects of cognitive, visual, and motor impairments on the outcomes of the study, it cannot be concluded that no confounding effects occurred. In future studies, including a control condition matched to the experimental paradigm where only the memory components are removed could better clarify this issue. Finally, considering the aim of this study, which was to develop a practical apparatus for WM assessment in clinical settings, we did not use an eye tracker. Although we attempted to minimize the effect of eye movement in both paradigms by instructing the participants to fixate on a central fixation point, using an eye tracker is necessary to validate these findings further.

In summary, using the resource-based model paradigms, we demonstrated that recall error and precision were impaired in MS. We provided new insight regarding the progressive state of the disease by assessing the plausible mechanisms related to the dissociable behavior of MGL and sequential paradigms in classifying MS subtypes. Furthermore, applying a computational model capable of disentangling error distribution in recalling information allowed us to uncover new insight regarding WM deficit in the MS population. Our results determined that decreased signal-to-noise ratio and swap error were responsible for WM deficit in MS. Overall, this study provided a sensitive measure for assessing WM impairment and gave new insight into the organization of WM dysfunction in MS.

## Materials and methods
### Participants
A total of 121 patients with confirmed MS (61 RRMS and 60 SPMS) and 73 healthy control volunteers participated in this study. Patients were recruited in a full-census manner from the Comprehensive Multiple Sclerosis Clinic at Kashani Hospital in Isfahan between February 2021 and January 2022. The following criteria were used for participant selection: diagnosis based on the 2017 McDonald criteria (*Thompson et al., 2018*), age between 18–55 y, diagnosis within 1–18 y prior to entering the study, an expanded disability status scale (EDSS) score of 0–6.5, no history of clinical relapse or corticosteroid therapy within 2 mo before entering the study, normal or corrected-to-normal visual acuity and color vision (based on the recorded bedside neurological examination in their profile), and no impairment in factors that could interfere with the study including visual acuity, visual field, extraocular movement, scotoma, nystagmus, or tremor in the upper extremity, assessed as part of the EDSS examination. Additionally, normal performance in the Nine-Hole Peg Test (9-HPT < 45 s), the absence of a history of brain surgeries, major neurological disorders (stroke, epilepsy, brain tumor, or CNS infection), psychiatric disorders (major depressive disorder, bipolar disorder, or schizophrenia), drug or alcohol abuse, and chronic systemic disorders (diabetes, renal failure, liver failure, chronic obstructive pulmonary disorder [COPD], hypothyroidism, or hyperthyroidism) were required for participation in this study. Moreover, the control group should not have a family history of MS in their first-degree relatives.

### Procedure
The study began with a clinical interview and neurological examination to collect clinical and demographic information from the participants. Participants also completed 9-HPT and the Persian version of the MoCA test (*Badrkhahan et al., 2020*; *Goodkin et al., 1988*; *Nasreddine et al., 2005*). MoCA is a standard cognitive screening tool with a scoring system ranging from 0 to 30, where distinct ranges correspond to different levels of cognitive function. Scores of ≥26 indicate normal cognitive performance, 18–25 denote mild cognitive impairment, 10–17 determine moderate cognitive impairment, and ≤10 is considered severe impairment.

Participants received verbal and written instructions before beginning the assessment. A 10-trial training block was then performed to ensure their understanding of the tasks. After this, the actual assessment started.

Written informed consent and consent for publication have been obtained from all participants before the start of the study. This study followed the latest update of the Declaration of Helsinki (*World Medical Association, 2013*) and was approved by the Iranian National Committee of Ethics in Biomedical Research (approval ID: IR.MUI.MED.REC.1400.441).

### Visual working memory paradigms
Visual WM was assessed using two analog recall tasks, MGL and analog recall paradigms with sequential bar presentation. Stimuli were presented on a 15″ cathode ray tube (CRT, 75 Hz refresh rate) monitor at a distancing view of 48 cm. The paradigms were run in a dimly lit room on a computer with a Linux operating system and MATLAB software (MATLAB 2019a, The MathWorks, Inc, Natick, MA) with Psychtoolbox 3 extension (*Brainard, 1997*; *Pelli, 1997*).

#### Memory-guided localization
Each trial was initiated by presenting a central fixation point (diameter of 0.51°) for 2 s, followed by the presentation of a target (a filled green circle with a diameter of 1.29°) for 1 s. The target randomly

appeared at different eccentricities (3.22°, 6.44°, or 9.66°) on each trial. In each block, targets were presented with equal probability at each eccentricity in random order (pseudo-random selection). While encouraging participants to maintain fixation on the central fixation point, participants were asked to memorize the location of the target circle for a delay period of 0.5, 1, 2, 4, or 8 s (chosen pseudo-randomly). After the delay period, the fixation point changed from a circle to a cross, indicating the end of the delay period. Participants were asked to locate the target's position using the computer mouse and confirm their response by pressing the left button on the mouse. Subsequently, visual feedback was presented, showing them the correct position of the target and their response (*Figure 1A*). Participants completed six blocks of 30 trials. They also completed a 10-trial training block before the start of the study. Recall error (absolute error), Euclidian distance between the target's location and subject response in visual degree, and RT were recorded for further assessment.

## Sequential paradigm with bar stimuli

Two designs of analog report tasks with sequential bar presentation, that is, the low memory load condition (1 bar) and high memory load condition (3 bar), were developed to evaluate the visual WM deficit in MS. In the high memory load condition, each trial started with a small central fixation point (0.26°) for 2 s, followed by a sequence of three distinguishable colored bars (red, green, and blue) at the center of the screen in a pseudo-random order. Each bar (2.57° by 0.19°) was presented for 500 ms, followed by a 500 ms blank interval. The minimum angular difference between the consecutively presented bars was 10°. Participants were asked to memorize both the orientation and color of the presented bars. After the bars were presented, a single bar, the 'probe bar' cued with the color of one of the presented bars, was displayed. Participants were asked to adjust the orientation of the probe bar, presented vertically, to match the orientation of the bar with the same color (target bar). To do that, they used a computer mouse and confirmed their response by clicking the right button. They received visual feedback, which showed the correct orientation of the target bar, their response, and the difference between them in angular degree (*Figure 1B*). The high memory load condition consisted of six blocks, each with 30 trials. The low memory load condition had the same structure as the high memory load condition except for presenting one bar instead of three (*Figure 1C*). After the high memory load condition, subjects participated in 30 trials of the low memory load condition. Due to the 1 bar design of low memory load condition, the swap error was absent, so fewer trials were needed (*Peich et al., 2013*). Before starting the paradigm, they also participated in a 10-trial training block with the same structure as the low memory load condition. The orientation of presented bars, subject response, recall error (absolute error), angular difference between the target value and subject response, and RT were recorded.

## Statistical analysis

Statistical analyses were conducted using IBM SPSS Statistics for Mac, version 26 (IBM Corp., Armonk, NY). The values were reported as mean ± standard deviation (SD). Data with extreme outliers (values greater than third quartile +3× interquartile range or less than first quartile – 3× interquartile range) in MGL and sequential paradigm with 3 bar were excluded from further analysis. The level of significance was set at p-value<0.05.

Clinical and demographic profiles of the participants, except for the gender, cognitive ability, and treatment history, were compared using one-way ANOVA or Kruskal–Wallis *H* test (three groups comparison) and independent-sample *t*-test or Mann–Whitney *U* test (two groups comparison). The post hoc Tukey's and Dunn's multiple-comparison tests were performed following the significant results of ANOVA and Kruskal–Wallis *H* test. Also, Bonferroni correction for multiple tests was performed following Dunn's post hoc analysis, and the adjusted p-value was reported.

The cognitive ability of participants was classified according to the mentioned ranges of MoCA score. Patients' disease-modifying therapy (DMT) was classified into platform and non-platform treatments. In our study, platform treatments include interferon β-1a and glatiramer acetate, and non-platform treatments contain rituximab, ocrelizumab, fingolimod, dimethyl fumarate, and natalizumab. These categorical variables, along with gender, were compared using the chi-square test.

For the MGL paradigm, recall error, recall precision (defined as the reciprocal of the standard deviation of recall error), and RT were compared between groups (healthy, RRMS, and SPMS) among different conditions (distance or delay, mixed model ANOVA, between- and within-subjects

comparisons). For the sequential paradigm, since the data was in a circular space, based on the method proposed by *Fisher, 1993*, we used the analog report MATLAB toolbox of Bays Lab (*Bays Lab, 2020*) to calculate the circular mean of recall error for each subject. Similarly, recall precision, defined as the reciprocal of the circular standard deviation of recall error, was calculated for each subject. For both high and low memory load conditions (3 bar and 1 bar), recall error, recall precision, and RT were compared between groups with respect to the order of the presented bars (mixed model and one-way ANOVA). To further investigate the sources of error in recalling information and uncover the involved mechanisms in visual WM impairment, the Mixture Model, a probabilistic model developed before (*Bays et al., 2009*; *Schneegans and Bays, 2016*), was utilized. The Mixture Model considers three possible sources for information recall. They are defined as the Gaussian variability in reporting the target and non-target values and random responses (*Bays et al., 2009*; *Schneegans and Bays, 2016*). In our study, they referred to reporting the orientation of the target bar (target proportion), misreporting the orientation of the other two non-target bars instead of the target bar (swap error), and random response (uniform proportion). Using the Mixture Model, the probabilities of target, non-target, and uniform responses and the von Mises distribution concentration score were calculated for each subject. The sources of error were evaluated by comparing the probability of target, non-target, and uniform responses between groups among different bar orders (mixed model ANOVA). Also, von Mises SD, defined as the circular standard deviation of the concentration score of von Mises distribution, was assessed using the same method. Moreover, nearest-neighbor analysis was performed to further evaluate the isolated effect of orientation (*Pertzov et al., 2013*). In this set of analyses, we removed the effect of swap error by defining recall error as the difference between the subject response and the nearest presented bar in each trial. The isolated effect of orientation was assessed between groups among different bar orders (mixed model ANOVA). Finally, due to the 1 bar design of the low memory load condition, swap error was not present; hence, only the target proportion, uniform proportion, and von Mises SD were compared (one-way ANOVA).

For each comparison, hierarchical regression analyses were performed to evaluate the possible confounding effect of demographic and clinical variables (significantly different between groups) on results. Finally, the dissociable function of MGL and sequential paradigms in distinguishing healthy control from MS patients, healthy control from MS subtypes, and MS subtypes from each other were assessed by performing ROC analysis. Recall error was used for classification purposes, and the area under the curve (AUC) was calculated as a measure of the paradigms' accuracy in group classification.

## Acknowledgements

We sincerely thank Farinaz Tabibian, Leila Sadat Razian, Maryam Mokhtari, and Zahra Mohamad Hoseiny for their cooperation in coordinating patients, Shahrzad Mohammadpour for advice about model fitting, and Mohammadamin Motaharinia for help regarding programming issues and his comments on the first draft of this manuscript. This study was supported by the Isfahan University of Medical Sciences (grant number: 2400104).

---

## Additional information

### Funding

| Funder | Grant reference number | Author |
|---|---|---|
| Isfahan University of Medical Sciences | Startup Grant to Center for Translational Neuroscience | Vahid Shaygannejad Fereshteh Ashtari Iman Adibi |
| Isfahan University of Medical Sciences | 2400104 | Vahid Shaygannejad Fereshteh Ashtari Iman Adibi |

The funders had no role in study design, data collection and interpretation, or the decision to submit the work for publication.

## Author contributions
Ali Motahharynia, Conceptualization, Data curation, Software, Formal analysis, Investigation, Visualization, Methodology, Writing – original draft, Writing – review and editing; Ahmad Pourmohammadi, Conceptualization, Data curation, Software, Investigation, Methodology, Writing – review and editing; Armin Adibi, Investigation, Writing – review and editing; Vahid Shaygannejad, Fereshteh Ashtari, Resources, Funding acquisition, Writing – review and editing; Iman Adibi, Conceptualization, Resources, Formal analysis, Supervision, Funding acquisition, Methodology, Writing – review and editing; Mehdi Sanayei, Conceptualization, Software, Formal analysis, Supervision, Methodology, Writing – review and editing

## Author ORCIDs
Ali Motahharynia https://orcid.org/0000-0002-1140-3257
Ahmad Pourmohammadi http://orcid.org/0000-0002-4996-0569
Iman Adibi http://orcid.org/0000-0003-4409-9404
Mehdi Sanayei http://orcid.org/0000-0003-3593-8018

## Ethics
Written informed consent and consent for publication have been obtained from all participants before the start of the study. This study followed the latest update of the Declaration of Helsinki and was approved by the Iranian National Committee of Ethics in Biomedical Research (Approval ID: IR.MUI. MED.REC.1400.441).

Reviewer #1 (Public Review): https://doi.org/10.7554/eLife.87442.3.sa1
Reviewer #2 (Public Review): https://doi.org/10.7554/eLife.87442.3.sa2
Author Response https://doi.org/10.7554/eLife.87442.3.sa3

---

# Additional files

## Supplementary files
• Supplementary file 1. Hierarchical regression analysis for the MGL paradigm.

• Supplementary file 2. Statistical results of reaction time for the MGL paradigm.

• Supplementary file 3. Hierarchical regression analysis for the sequential paradigm with 3 bar, high memory load condition.

• Supplementary file 4. Statistical results of reaction time for the sequential paradigms (3 bar and 1 bar).

• Supplementary file 5. Hierarchical regression analysis for the sequential paradigm with 1 bar, low memory load condition.

• MDAR checklist

• Source data 1. Psychophysics data of participants in the MGL paradigm.

• Source data 2. Psychophysics data of participants in the sequential paradigm with 3 bar.

• Source data 3. Psychophysics data of participants in the sequential paradigm with 1 bar.

## Data availability
All data generated or analyzed during this study are included in the manuscript and supporting files; Source Data files have been provided for psychophysics data, Figures 2, 3, and 4, Figure 3-figure supplement 1, and Tables 1 and 2. The source codes for the analog report MATLAB toolbox (*Bays Lab, 2020*) are available here.

---

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
