## [Editor Report · eLife assessment]

This paper provides **valuable** information regarding visuospatial working memory performance in patients with MS compared to healthy controls, using a relatively novel continuous measure of visual working memory. There are some weaknesses with the way the clinical groups were matched, but those limitations are acknowledged and the strength of evidence for the claims is otherwise **convincing**. The paper will be of interest to those working in the field of clinical neuroscience.

---

## [Referee Report · Reviewer #1 (Public Review)]

This study compares visuospatial working (VWM) memory performance between patients with MS and healthy controls, assessed using analog report tasks that provide continuous measures of recall error. The aim is to advance on previous studies of VWM in MS that have used binary (correct/incorrect) measures of recall, such as from change detection tasks, that are not sensitive to the resolution with which features can be recalled, and to use mixture modelling to disentangle different contributions to overall performance. The results identify a specific decrease in the precision of VWM recall in MS, although the possibility that visual and/or motor impairments contribute to performance decrements on the memory task cannot be ruled out.

---

## [Referee Report · Reviewer #2 (Public Review)]

The authors applied two visual working memory tasks, a memory-guided localization (MGL), examining short-term memory of the location of an item over a brief interval, and an N-back task, examining orientation of a centrally presented item, in order to test working memory performance in patients with multiple sclerosis (including a subgroup with relapsing-remitting and one with secondary progressive MS), compared with healthy control subjects. The authors used an approach in testing and statistically modelling visual working memory paradigm previously developed by Paul Bays, Masud Husain and colleagues. Such continuous measure approaches make it possible to quantify the precision, or resolution, of working memory, as opposed to measuring working memory using discretised, all-or-none measures. This represents an advance beyond prior work in this area.

The authors of the present study found that both MS subgroups performed worse than controls on the N-back task and that only the secondary progressive MS subgroup was significantly impaired on the MGL task. The underlying sources of error including incorrect association of an object's identity with its location or serial order, were also examined.

The application of more precise psychophysiological methods to test visual working memory in multiple sclerosis should be applauded. It has the potential to lead to more sensitive and specific tests which could potentially be used as useful outcome measures in clinical trials of disease-modifying drugs, for example.

The present study does not compare the continuous-report testing with a discrete measure task so it is unclear whether the former is more sensitive, or more feasible in this patient group, although this may not have been the purpose of the study.

Comments on the revised submission: My previous comments have been answered to the extent that is possible with the data available.

---

## [Author Response]

The following is the authors’ response to the current reviews.

**eLife assessment**
This paper provides valuable information regarding visuospatial working memory performance in patients with MS compared to healthy controls, using a relatively novel continuous measure of visual working memory. There are some weaknesses with the way the clinical groups were matched, but those limitations are acknowledged and the strength of evidence for the claims is otherwise convincing. The paper will be of interest to those working in the field of clinical neuroscience.

We are grateful to the editors and reviewers for their careful review of our manuscript and their dedicated time and effort. Their valuable feedback has been instrumental in improving the quality of our work.

**Reviewer #1 (Public Review):**
This study compares visuospatial working (VWM) memory performance between patients with MS and healthy controls, assessed using analog report tasks that provide continuous measures of recall error. The aim is to advance on previous studies of VWM in MS that have used binary (correct/incorrect) measures of recall, such as from change detection tasks, that are not sensitive to the resolution with which features can be recalled, and to use mixture modelling to disentangle different contributions to overall performance. The results identify a specific decrease in the precision of VWM recall in MS, although the possibility that visual and/or motor impairments contribute to performance decrements on the memory task cannot be ruled out.

Although we try to address this matter by clinical screening, as the reviewer mentioned, the possibility that visual and/or motor impairments contribute to performance decrements on the memory task cannot be ruled out. Therefore, in future studies, including a control condition matched to the experimental paradigm where only the memory components are removed is needed to elucidate this issue.

**Reviewer #2 (Public Review):**
The authors applied two visual working memory tasks, a memory-guided localization (MGL), examining short-term memory of the location of an item over a brief interval, and an N-back task, examining orientation of a centrally presented item, in order to test working memory performance in patients with multiple sclerosis (including a subgroup with relapsing-remitting and one with secondary progressive MS), compared with healthy control subjects. The authors used an approach in testing and statistically modelling visual working memory paradigm previously developed by Paul Bays, Masud Husain and colleagues. Such continuous measure approaches make it possible to quantify the precision, or resolution, of working memory, as opposed to measuring working memory using discretised, all-or-none measures. This represents an advance beyond prior work in this area.The authors of the present study found that both MS subgroups performed worse than controls on the N-back task and that only the secondary progressive MS subgroup was significantly impaired on the MGL task. The underlying sources of error including incorrect association of an object's identity with its location or serial order, were also examined. The application of more precise psychophysiological methods to test visual working memory in multiple sclerosis should be applauded. It has the potential to lead to more sensitive and specific tests which could potentially be used as useful outcome measures in clinical trials of disease-modifying drugs, for example. The present study does not compare the continuous-report testing with a discrete measure task so it is unclear whether the former is more sensitive, or more feasible in this patient group, although this may not have been the purpose of the study.

The reviewer brought up an important point, but as they stated, it was not the focus of our current study. Nevertheless, it is a valuable suggestion for future research to compare continuous with discrete measure paradigms to assess their sensitivity and feasibility in the MS population.

The following is the authors’ response to the original reviews.

We thank the editors and reviewers for their thorough reading of this manuscript and valuable suggestions. We appreciate the time and effort they have put into this manuscript to provide feedback for improving our work. Based on their comments, we carefully considered their suggestions and revised the manuscript to address their concerns. Our one-by-one response to reviewers comments is as follows.

**Reviewer #1 (Public Review):**
This study compares visuospatial working memory performance between patients with MS and healthy controls, assessed using analog report tasks that provide continuous measures of recall error. The aim is to advance on previous studies of VWM in MS that have used binary (correct/incorrect) measures of recall, such as from change detection tasks, that are not sensitive to the resolution with which features can be recalled, and to use mixture modelling to potentially disentangle different contributions to overall performance. This aim is met in part, but there are some problems with the authors' interpretation of their findings:1. How can the authors be confident the performance deficits in the patient groups are impairments of working memory and not visual or motor in nature? I appreciate there was some kind of clinical screening, but it seems like there should have been a control condition matched to the experimental tasks with only the memory components removed.

We appreciate the reviewer’s concern regarding the potential confounding effects of visual or motor impairment on the outcomes of our study.

While we acknowledge that a control condition with only the memory components removed could have further strengthened our results, we did not include one, which is a limitation of the current study.

To address this limitation, we conducted clinical screening to ensure that the observed deficit was due to working memory impairment and not visual or motor in nature. As part of the expanded disability status scale (EDSS) evaluation, we did not include individuals with issues such as visual acuity, visual field, and extraocular movement impairment, scotoma, nystagmus, and tremors in the upper extremity, which could interfere with the study. Moreover, participants were screened using the 9-Hole Peg Test (9-HPT) before entering the study. These evaluations helped us to ensure that participants with impaired visual or motor performance, which could potentially confound the study, were not included. Our effort to remove the confounding factors with clinical screening provided additional insight into the interpretability of the results. We have updated our inclusion/exclusion criteria accordingly and included this limitation in our discussion.

2. The participant groups are large, which is definitely a strength, but not particularly well-matched in terms of demographics, with notable differences in age (mean and spread), years of education and gender. These could potentially contribute to differences in performance between groups and tasks.

We appreciate the reviewer's comment and agree that a matched control group would be ideal. However, we addressed this issue using hierarchical regression analysis.

Our study assessed visual working memory (VWM) resolution using two analog recall paradigms: the sequential paradigm with bar stimuli and memory-guided localization (MGL). While the demographic data of gender, age, and education in the MGL paradigm were matched between patients and control group, there was a significant difference in these factors between groups in the sequential paradigm.

To address this issue, we performed hierarchical regression analysis to compare recall parameters in the sequential paradigm with 3-bar and 1-bar stimuli, respectively. We assessed for the confounding effect of gender, age, and education, and the results were presented in supplementary tables 3 and 5.

In the sequential paradigm with 3-bar stimuli (high memory load condition), we found that all recall parameters were significantly different between groups. However, after adjusting for age and education, the result became insignificant for uniform response proportion. In the 1-bar paradigm (low memory load condition), while the results were significantly different between groups, after adjusting for gender, age, and education, target and uniform response proportions became insignificant (uniform proportion = 1 – target proportion, since there was no swap error in the 1-bar condition).

3. The authors interpret the mixture model parameter described as "misbinding error" as reflecting failures of feature binding, and propose a link to hippocampus on that basis, however there is now quite strong evidence that these errors (often called swaps) are explained mostly or entirely by imprecision in memory for the cue feature (bar color in this case), e.g. McMaster et al. (2022), already cited in the ms.

We thank the reviewer for this valuable comment regarding interpreting the mixture model parameter, described as a “misbinding error” in our study.

Swap error has been attributed to different mechanisms, including the variability in cue feature dimension, cue-independent sources, and strategic guessing. As the reviewer mentioned, in a recent study by McMaster et al., a comprehensive evaluation of these hypotheses was performed and determined that the variability in cue feature dimension could solely explain the occurrence of swap error.

In response to this comment, we have added a discussion of this matter, the neural correlates of swap error, and the possible explanation for this phenomenon in multiple sclerosis (MS) population to the seventh paragraph of the discussion. Additionally, since our study did not include neuroimaging assessment, we have discussed the results from neuroanatomical points of view to further explain the possible structures involved in the occurrence of swap errors in MS. The seventh and eighth paragraphs of the discussion have been revised for further clarification.

4. The methodology of the ROC analyses should be described in more detail: it is not clear what measures are being used to classify participants or how.

This matter is clarified in the results and the last paragraph of materials and methods. In both paradigms, recall error was used for classification purposes.

5. There are a number of unusual choices of terminology that could potentially confuse or mislead the reader: The tasks are not "n-Back" tasks by the usual meaning: they are analog report tasks with sequential presentation. The terms recall "error", "variability", "precision" and "fidelity" are used idiosyncratically. Variability and precision usually refer to the same thing: they describe the dispersion or spread of errors. The measure described as recall error in the sequential tasks is presumably absolute (or unsigned) error. For the mixture model parameters I suggest describing them more explicitly in terms of the mixture attributes, e.g. "Von Mises SD", "Target proportion", "Non-target proportion" "Uniform proportion".

We thank the reviewer for this suggestion. We have made revisions to clarify the terminology used in our study.

The term "n-back" is changed to an analog recall paradigm with sequential presentation. Additionally, as mentioned in the materials and methods, the recall error in the MGL paradigm is the Euclidian distance between the target's location and subject response in visual degree. In the sequential paradigms, this value is the angular difference between the response and target value, in which both are absolute errors. To avoid confusion, we have added the term "absolute error" alongside the term "recall error" to provide a clear understanding of this measurement. Moreover, as the reviewer suggested, we changed "recall variability" to "von Mises SD" for a more precise description. We also changed the remaining terms to "target proportion", "swap error (non-target proportion)", and "uniform proportion".

**Reviewer #2 (Public Review):**
The authors applied two visual working memory tasks, a memory-guided localization (MGL), examining short-term memory of the location of an item over a brief interval, and an N-back task, examining orientation of a centrally presented item, in order to test working memory performance in patients with multiple sclerosis (including a subgroup with relapsing-remitting and one with secondary progressive MS), compared with healthy control subjects. The authors used an approach in testing and statistically modelling visual working memory paradigm previously developed by Paul Bays, Masud Husain and colleagues. Such continuous measure approaches make it possible to quantify the precision, or resolution, of working memory, as opposed to measuring working memory using discretised, all-or-none measures.The authors of the present study found that both MS subgroups performed worse than controls on the N-back task and that only the secondary progressive MS subgroup was significantly impaired on the MGL task. The underlying sources of error including incorrect association of an object's identity with its location or serial order, were also examined.The application of more precise psychophysiological methods to test visual working memory in multiple sclerosis should be applauded. It has the potential to lead to more sensitive and specific tests which could potentially be used as useful outcome measures in clinical trials of disease modifying drugs, for example.However, there are some significant limitations which severely affect the scientific validity and interpretability of the study:1. There is a striking lack of key clinical information:(1.1) The inclusion and exclusion criteria are unclear and a recruitment flowchart has not been provided. Therefore it is unclear what proportion of MS patients were ineligible due to, for example, visual impairment.

We thank the reviewer for raising this matter. To address this issue, we revised the first section of materials and methods to include detailed inclusion/exclusion criteria information. However, it is important to note that we recruited the patients in a full-census manner, where only the patients who fulfilled the inclusion criteria participated. Unfortunately, we did not record the number of patients who did not meet the inclusion criteria.

(1.2) Basic clinical data such as EDSS scores, disease duration, treatment history, and performance on standard cognitive testing were not provided. Basic clinical and demographic data for each subgroup were not provided in a clear format. This severely limits the interpretability of the study and its significance for this clinical population. For example, might it be that the SPMS patients performed worse on the MGL task because they were more cognitively impaired than RRMS patients? That question might be easily answered, but the answer is unclear based on the data provided.

We appreciate the reviewer for bringing up this important concern. To further clarify the basic clinical and demographic data, we have revised tables 1 and 2 to include detailed information regarding gender, age, education, cognitive ability, disease duration, EDSS score, and disease-modifying therapy (DMT) for each group, respectively. The information is reported as mean ± standard deviation except for the categorical data.

Regarding the participants' cognitive ability, we added the Montreal cognitive assessment test results for both paradigms. MoCA is a standard cognitive screening tool that has a score of 0 to 30. The different ranges of MoCA scores related to the different levels of cognitive function, in which a score ≥ 26 is considered normal cognitive ability, 18-25 denotes mild cognitive impairment, 10-17 determines moderate cognitive impairment, and a score ≤ 10 is considered severe impairment.

First, we classify the participants based on their MoCA value and compare groups with each other. While the primary results showed that patient groups were more impaired compared to healthy controls, our results remained significant after adjusting for MoCA using hierarchical regression analysis. This suggests that the observed difference was not solely due to more cognitive impairment in the patients' population.

Moreover, the information regarding the treatment history of patients is added in the following format. DMT is classified into two groups, i.e., platform and non-platform treatments. In our study, the platform treatments include interferon beta-1a and glatiramer acetate, and non-platform treatments include rituximab, ocrelizumab, fingolimod, dimethyl fumarate, and natalizumab. In both paradigms, the patients did not significantly differ based on the received therapy.The MoCA assessment and treatment history information is added to tables 1 and 2 and supplementary tables 1, 3, and 5. Moreover, the second paragraph of materials and methods, second paragraph of statistical analysis in materials and methods, and the appropriate sections of the results are revised.

2. The study is completely agnostic to the underlying pathophysiology. There is no neuroimaging available, therefore it is unclear how the specific working memory impairments observed might relate to lesioned underlying brain networks which are crucial for specific aspects of working memory. This severely limits the scientific impact of the results. This limitation is acknowledged by the authors, but the authors did not put forward any hypotheses on how their results may be underpinned by the underlying disease processes.

We appreciate the reviewer for this valuable suggestion. To further strengthen the connection between our findings and the possible underlying mechanisms of WM dysfunction in MS, we have added a discussion from the neuroanatomical perspective in the eighth paragraph of the discussion section.

3. The present study does not compare the continuous-report testing with a discrete measure task so it is unclear if the former is more sensitive, or more feasible in this patient group, although this may not have been the purpose of the study.

The reviewer pointed out an interesting matter. However, this was not the focus of the current study. Nonetheless, it is a valuable suggestion for future work to compare continuous vs. discrete measure paradigms to determine their sensitivity and feasibility in the MS population.